Thermal-bias PCR: generation of amplicon libraries without degenerate primer interference

http://orcid.org/0000-0003-1606-7771 Moore Sean D. 1 2 sean.moore@ucf.edu
1 Burnett School of Biomedical Sciences, University of Central Florida , Orlando, Florida , United States
2 Genomics and Bioinformatics Cluster, University of Central Florida , Orlando, Florida , United States
Sotelo-Mundo Rogerio
Electronic publication date: 2025 Oct 24
Publication date: 2025
Volume: 13
Electronic Location ID: e20241
Received 2025 Jul 16; Accepted 2025 Sep 24
Copyright: © 2025 Moore
Copyright year: 2025
Copyright holder: Moore
License: This is an open access article distributed under the terms of the Creative Commons Attribution License, which permits unrestricted use, distribution, reproduction and adaptation in any medium and for any purpose provided that it is properly attributed. For attribution, the original author(s), title, publication source (PeerJ) and either DOI or URL of the article must be cited.
License URL: https://creativecommons.org/licenses/by/4.0/

Keywords: Polymerase chain reaction, qPCR, Sequencing library, Degenerate primers, PCR model

Funding: The author received no funding for this work.

==============================
The polymerase chain reaction (PCR) has been used to amplify specific gene regions for many taxonomic studies and there have been substantial efforts to develop protocols that efficiently amplify target regions from a majority of mixed-template populations. Most protocols include the use of degenerate oligonucleotide primer pools, which contain mixed nucleotide sequences to improve priming from templates containing non-consensus sequence variations in their primer-binding sites. In this work, computational modeling and experimental measurements revealed that degenerate primers reduce efficiency well before a substantial product pool has been generated. It was also discovered that non-degenerate primers produced amplicons significantly better than their degenerate counterparts when amplifying either a consensus or a non-consensus target. Using quantitative, real-time PCR (qPCR) and data fitting as a guide, a new PCR protocol was developed that avoids the use of degenerate primers and allows for the stable amplification of targets containing mismatches to the targeting primers. This protocol involves the use of only two non-degenerate primers with no intermediate processing steps and it allows for the reproducible production of amplicon sequencing libraries that maintain the fractional representations of rare members.

Introduction

Polymerase chain reaction (PCR) is used to generate deep sequencing libraries from mixed genome samples by targeting segments of conserved genes that reveal evolutionary relatedness (Gohl et al., 2016). When a cohort of organisms is identified that naturally contain primer-target mismatches relative to the consensus, a common strategy to increase representation in amplicon libraries is to introduce sequence heterogeneity into the oligonucleotide pools (so-called ‘primer degeneracy’) such that there are perfectly matched primers available (Campos, Gallardo & Quesada, 2023). While the intent of including primer degeneracy is to increase non-consensus target priming, the degeneracy can negatively impact the amplification of the entire amplicon pool, including prevalent consensus targets (Gaby & Buckley, 2017). Thus, increasing rare-target representation by this approach comes with the cost of suppressing, and potentially distorting, the representation of other targets (Sipos et al., 2007; Bonk et al., 2018). PCRs used to generate sequencing libraries are not monitored using qPCR because there is no need to reevaluate template abundance at that stage. In this work, it is shown that global fitting of quantitative, real-time PCR (qPCR) data obtained during sequencing library generation revealed a previously unrecognized metric that can be used to evaluate reaction quality and primer performance. Using that metric as a guide, a new single-reaction PCR method was developed that allows for the formation of deep sequencing libraries that accurately reveal changes in community structure.

A focus was placed on the amplification of segments of the V3-V4-V5 variable regions of bacterial genes encoding the 16S rRNA (rrs) because of their extensive use in the characterization of complex bacterial communities. Each region is flanked by highly-conserved sequences, which have been targeted by a variety of primer pairs over several decades (Campos, Gallardo & Quesada, 2023; Baker, Smith & Cowan, 2003). Although these primer binding sites are highly conserved, they are not universal, so multiple iterations of primer redesign have been implemented to allow the primers to anneal to a wider variety of non-canonical 16S rRNA gene targets. In most cases, such primers were altered by introducing sequence degeneracy at defined positions. Commonly employed degenerate primer pairs that target the eubacterial V3-V4 segment of the 16S rRNA gene were designed by Klindworth et al. (2013), and a more degenerate pair that additionally targets Archaea was designed by Takahashi et al. (2014). A degenerate pair initially designed by Caporaso et al. (2011) targets the V4 region, and those were made more degenerate by Apprill et al. (2015) to better cover marine samples. The V4-V5 region can be amplified using a degenerate pair designed by Quince et al. (2011) and Parada, Needham & Fuhrman (2016). While computational assessments of sequence heterogeneity at primer targeting sites clearly reveal variations among distantly-related bacteria and mismatches lead to underrepresentation in libraries (Eloe-Fadrosh et al., 2016), degenerate primer designs are guided by the idea that mismatched primers cannot function well and that degenerate primers improve PCR performance (Bru, Martin-Laurent & Philippot, 2008).

The overall number of different primer sequences in degenerate pools can be extensive (e.g., the Takahashi pair contains 36 different sequences). Because primer concentrations in PCRs are typically ~107 times higher than the initial targets, having extensive primer diversity may seem unimportant; however, mismatched primers anneal at low temperatures, yet they may not be tolerated by the polymerase and act as reaction inhibitors. When mismatched primers are incorporated into amplicons, they convert the primer binding sites to sequences that unpredictably bias subsequent priming (Gaby & Buckley, 2017). Added to this scenario is the progressive depletion of functional primers because incorporation of best-matching oligonucleotides in early rounds is favored. To address these issues, alternative protocols that separate a degenerate template-targeting stage from a non-degenerate library amplification stage have been developed (Green, Venkatramanan & Naqib, 2015; Naqib, Poggi & Green, 2019; Naqib et al., 2019; Kahsen et al., 2024). Those procedures require cleaning intermediate samples and mixing new reactions, which adds substantial labor and reagent costs, especially for larger projects.

In qPCR assays, the cycle at which a reaction curve crosses a pre-established threshold is used as an indicator of a positive or negative test result; whereas in other cases, changes in ΔCq between a reference and a target are used either to quantify a target or to measure changes in target abundance. In each case, these methods rely on measuring changes to one numerical value (Cq) that is reliably indicative of abundance (Bustin et al., 2009; Bustin & Huggett, 2017; Ruiz-Villalba, Ruijter & Van Den Hoff, 2021). In prior work, we developed a mathematical PCR model that uses two variables to define the shape of a PCR amplification profile (Carr & Moore, 2012). Computational fitting of qPCR data using this model provides unique values for each reaction. Once obtained, those two values allow for a calculation of the amount of initial template (Carr & Moore, 2012). Relative differences in template abundance obtained by this method mirror the differences obtained by cycle-threshold analysis (ΔCq) of the same data, but global fitting results typically exhibit less variance among technical replicates.

Inspections of global fitting data from various qPCR experiments revealed that the ratio of those two global fitting values was apparently indicative of overall reaction quality. In this report, I describe the use of that ratio as a dimensionless metric to evaluate reaction quality among samples amplified under different conditions, with lower ratios indicating higher quality reactions. This ratio allowed for the interrogation of primer performance during sequencing library preparations, which revealed that final amplicon yield is an unreliable measure of reaction quality, and that degenerate primers substantially reduce reaction performance. Engineered reporter templates were used to monitor amplicon production using non-degenerate primers in an effort to improve performance on mismatched targets. From those studies, a “thermal-bias” PCR protocol was developed that uses only two non-degenerate primers in a single reaction by exploiting a large difference in annealing temperatures to isolate the targeting and amplification stages. Thermal-bias PCR allows for a proportional amplification of targets containing substantial mismatches in their primer binding sites and it can be used to generate deep sequencing libraries from mixed genome samples.

Materials and Methods

Reaction modeling

Modeled PCR data were generated using Excel (Microsoft). Input cells referenced for the model included: “seed”, “max”, and “KD”. A “Maximum efficiency” reference cell converted a desired reaction efficiency to a yield relative to a 2-fold amplification in the model (=(efficiency * 2) −1 ), such that an efficiency input of 1.0 generated a value of 1.0, and an efficiency input of 0.98 generated a value of 0.96 (98% of 2-fold). The yield of the first cycle referenced the seed input cell, and the yields of subsequent cycles referenced the value in the cycle preceding them. An example of a model cell is “=B6*($L$9+(($G$6-B6)/$G$6)-(B6/($J$6+B6)))”, where B6 is the preceding cell, $L$9 is the efficiency cell, $G$6 is the max cell, and $J$6 is the KD cell.

PCR template preparations

Escherichia coli genomic DNA was purchased from Fisher Scientific (cat. AAJ14380MA), resuspended in DNA buffer (10 mM Tris-Cl, 0.1 mM EDTA, pH 8.0) and quantified using UV absorbance (260 nm). Serial dilutions were prepared in DNA buffer. A mock community bacterial genome mixture ‘ABRF-MGRG 10 Strain Even Mix Genomic Material’ was purchased from the American Type Culture Collection (ATCC, cat. MSA-3001) (Foox et al., 2021). It contains a mixture of Bacillus subitilis (Bs), Chromobacter violaceum (Cv), Escherichia coli (Ec), Entercoccus faecalis (Ef), Halobacillus halophilus (Hh), Haloferax volcanii (Hv), Micrococcus luteus (Ml), Pseudomonas fluorescens (Pf), Pseudoalteromonas haloplanktis (Ph), and Staphylococcus epidermidis (Se) genomes. This mixture is listed by the ATCC as having an even mixture of each genome and a company representative indicated that it contains an equal weight of each. However, the creators of this mixture prepared it as having an even concentration of each genome and they subsequently determined that the genomes may not be evenly represented (Foox et al., 2021). The match and mismatch reporter templates were generated using PCR with NEBNext Ultra II Q5 polymerase (New England Biolabs, cat. M0544S) by amplifying segments of the E. coli V3-V4 region of the 16S rRNA gene using primers that introduced restriction sites as insertions and also modified the targeting primer-binding sites in the mismatch version. Overlap extension PCR (Q5 polymerase) was then used to fuse the two segments. The final products were gel purified and the concentrations were determined by UV absorbance in a SynergyMX spectrophotometer (Biotek Instruments) with a sample diluted 10-fold in TE buffer in a 1 cm cuvette. Yields were ~0.2 μg/μL and had a 260/280 ratio of 1.85. Templates were then normalized to 0.1 μg/μL (0.31 μM), and stored in DNA buffer.

Primer and protocol performance assays

E. coli genomic DNA was diluted to 1.95 μg/mL and 1 μL was used per 20 μL of reaction mixture (97.5 pg/μL final). Platinum II Taq and Ultra II Q5 reactions were supplemented with EvaGreen dye (Jena Bioscience, cat. # PCR-379); iTaq Universal SYBR Green (Bio-Rad, cat. # 1725120) and SsoFast EvaGreen (Bio-Rad, cat.172-5201) were used without modification. The Earth Microbiome Project (EMP) ‘16S Illumina Amplicon Protocol’ uses a 45 s melt at 94 °C, 60 s anneal at 50 °C, and a 90 s extension at 72 °C (https://earthmicrobiome.org/protocols-and-standards/16s/) (Ul-Hasan et al., 2019; Gilbert, Jansson & Knight, 2014; Thompson et al., 2017). The evaluated thermal-bias protocols included an enzyme activation step (2 min at 98 °C), two targeting cycles (10 s melting at 98 °C and annealing/extension ranging from 2.5–10 min at 46–55 °C), and 30 or 40 amplification cycles (10 s melting at 98 °C and annealing/extension for 1 min ranging from 74–84 °C). Fluorescence measurements were taken at the end of each amplification cycle. These protocols included a melt-curve analysis at the end that progressed from 75–95 °C in 0.5 °C steps.

Primer sequences are listed in Data SD1 and were purchased from Eurofins Genomics as ‘NGS Grade’ and stored as 100 μM stocks in DNA buffer. Primer pairs were mixed in High-Performance Liquid Chromatography (HPLC)-grade water at 1.11 μM each and added to reactions at 45 % (0.5 μM final). Standard PCRs were performed in a Bio-Rad MJ Mini thermal cycler and qPCRs were performed in a Bio-Rad CFX96 Touch Real-Time system. For global fitting, fluorescence data were exported from the real-time system and saved in .csv format before being analyzed using an online qPCR fitting program (http://www.bioinformatics.org/ucfqpcr/) (Carr & Moore, 2012). The seed, max, and KD values were then transferred to Excel (Microsoft) for subsequent processing. Figure data were plotted using Prism 9 (Graphpad Software) and annotated using Illustrator (Adobe).

Reporter template amplification and restriction digestion

Conventional PCRs were performed using Platinum II Taq polymerase and qPCRs were performed using SsoFast EvaGreen unless otherwise indicated. The reporter templates were diluted 10,000-fold in DNA buffer and 1 μL was used per 20 μL of reaction mixtures. DNA from those PCRs was purified using an in-house PEG precipitation protocol: mixing with an equal volume of Polyethylene Glycol (PEG) precipitation reagent (20% PEG-8000, 1 M NaCl); incubated at room-temperature for 10 min; centrifuged at 14,000 RCF for 15 min; liquid removed; pellet washed once with 75% ethanol, centrifuged 10 min, liquid removed; washed once with 95% ethanol, centrifuged 5 min, liquid removed; air dried. DNA was resuspended in a volume of DNA buffer matching the original PCR reaction volumes. Separate gel electrophoresis evaluations of this PEG procedure indicated that a single round reduced a 2 μM spike of primers to a barely detectable level with a complete recovery of DNAs larger than ~300 bp. As with magnetic bead-adhesion protocols, the size cut-off of recovered DNAs can be increased by lowering the added fraction of PEG reagent.

Restriction digestions were performed at 37 °C for 120 min using either BamHI-HF or SpeI-HF (NEB, cat. R3136S and R3133S) according to the manufacturer’s instructions. In preliminary restriction digestion studies, it was noted that neither the BamHI or SpeI digestions went to completion. Suspecting there may have been contamination by templates lacking the restriction sites, BamHI- and SpeI-cleaved amplicon fragments were independently gel purified and then ligated back together to ensure the final templates contained the restriction sites prior to Q5 amplification. However, these preparations were also unable to be fully digested. Digestion kinetics were also evaluated over 24 h and using increased enzyme concentration and using a different lot of enzyme. Although the digestion progression was completed within 30 min, a small amount of residual undigested material persisted. Amplicon reannealing was performed by heating samples to 95 °C for 5 min in a thermal cycler and subsequent cooling to 25 °C.

Sequencing library preparation and analysis

Each 100 μL PCR mixture contained 2.5 μL of ATCC MSA-3001 genome mixture (~0.175 ng/μL final) and either 2.5 μL of DNA buffer (Library_1), a 1:1 reporter template mix at a 1/100,000 dilution (Library_2), or a 1/200,000 reporter dilution (Library_3). The thermal-bias primers TB_F_Tm55_v2 and TB_R_Tm55_v2 contained Illumina TruSeq adapter sequences and had calculated targeting Tms of 55 °C and amplification Tms of 80 °C. SsoFast EvaGreen mixtures were distributed into four 20 μL reaction wells and the thermal-bias protocol used two targeting cycles (annealed 5 min at 48 °C) and 30 amplification cycles (annealed 1 min at 78 °C). After cycling, reaction wells were pooled and the amplicons were purified using two rounds of PEG precipitation. Dried amplicons were resuspended in 60 μL of DNA buffer, quantified using a Quant-iT PicoGreen assay (Invitrogen), and normalized to 20 ng/μL. The normalized samples were evaluated using gel electrophoresis to confirm amplicon size and primer removal. Bar-coding and 250 nt Illumina sequencing was performed by a commercial service (Azenta Life Sciences).

Sequence reads were processed and mapped on a public Galaxy server (usegalazy.org) (Afgan et al., 2016): Cutadapt v5.0 (Martin, 2011) was used to remove reads having a quality scores less than 20 and lengths less than 200 nt, Bowtie2 v2.5.3 (Langmead & Salzberg, 2012) was used to map reads onto a reference FASTA file containing concatenated V3-V4 regions of each bacterium and the match and mismatch reporter sequences. Alignment maps and their indices were downloaded and visualized using JBrowse2 (Diesh et al., 2023), which was also used to determine read depths at each V3-V4 region.

Results

Primer degeneracy reduces PCR quality

We have used global fitting of qPCR data during other projects to establish relative template abundances (seed values) (Carr & Moore, 2012; Smith et al., 2019; Bose et al., 2021). Inspection of the associated max and KD values from those fitting operations revealed that the max/KD ratios for a given reaction were reflective of the amplification quality, with more ‘upright’ reaction curves (indicating more productive reactions) having lower ratios. With an interest in using max/KD ratios as a metric to assess reaction quality, modelling was used to establish the impact of varying max and KD terms on PCR profile shapes (Fig. S1). That exercise revealed that reaction ’poisoning’ is primarily responsible for inefficient amplification and that end-point yield measurements are unreliable indicators of reaction quality.

The amplification performance of degenerate primer pairs was evaluated by monitoring the accumulation V3-V4, V4-V5, or V4 amplicons from Escherichia coli genomic DNA under reaction conditions recommended by the Earth Microbiome Project (EMP). Primer names, sequences, and calculated Tms are listed in Data SD1. In each target case, the performance of the degenerate primer pair was considerably poorer than the non-degenerate pair, yielding higher max/KD ratios, lower overall yield, and lower seed values (Fig. 1). It is notable that the greatest performance disparity occurred during amplification of the V3-V4 region, in which the forward primer contained degeneracy at the −4 position relative to the 3′ end. It is well documented that polymerases are less tolerant of mismatches near the 3′ end (Bru, Martin-Laurent & Philippot, 2008; Wu, Hong & Liu, 2009; Gohl et al., 2021), and structural studies revealed that positions −1 through −4 interact with the DNA duplex binding region of polymerases and mismatches in those locations disrupt the active site (Johnson & Beese, 2004). Therefore, the location of this degenerate position likely exacerbated poor performance.

Figure 1 Degenerate primers impede amplification.

Amplicon production of E. coli 16S rRNA gene variable regions was monitored using qPCR. Non-degenerate and degenerate primer pairs were compared for each amplified region (four replicates each). These reactions used the EMP cycling protocol and Platinum II Taq polymerase (supplemented with EvaGreen reporter dye, 97.5 pg/ μ L genomic DNA, 0.5 μ M each primer). Primer pairs: V3-V4 (ND_F1 & ND_R1; Pro_341F & Pro_805R), V4-V5 (ND_V4_F & ND_V5_R; 515F-Y & 926R), V4 (ND_V4_F & ND_V4_R; 515F-Y & 806RB). Schematics of the degenerate primers and the positions of variations are indicated on the left of each panel. Each reaction’s data were globally fit to obtain seed, max, and KD values and then a max/KD ratio was calculated. Insets show averages with standard deviations in parentheses. The poorer performance of the degenerate primers was associated with higher max/KD ratios.

The relative performance of degenerate primers was also evaluated using other polymerases and reaction conditions. In each case, the degenerate pair performed poorly compared to the non-degenerate pair (Fig. S2). Importantly, even though the degenerate sets contained primers that perfectly matched the initial targets, the reaction performances were compromised before the amplicons substantially accumulated above the baselines, when less than 1% of the final amplicon pools had been generated. Therefore, limiting cycle numbers would not avoid this performance disparity.

Templates containing mismatches at priming sites

To evaluate primer performance on target templates containing mismatches to non-degenerate primers, an engineered V3-V4 reporter template was constructed that contained two ‘mismatch’ nucleotide changes in each of the forward and reverse primer binding sites of the target (Fig. 2A). These template changes are in positions predicted to be compensated by these degenerate primers (Fig. S3); yet, they represent a rather exotic case for the amplification of V3-V4 regions, as non-canonical eubacterial 16S rRNA genes from environmental samples usually contain only one or two mismatches to the consensus sequences at those positions, with rarer examples having highly deleterious mismatches farther into the 3’ primer-binding region (Gohl et al., 2021). A V3-V4 control template was also constructed that contained consensus ’match’ primer-binding sites. With an eye toward evaluating the amplification of mixed template samples, each engineered template additionally contained a unique restriction enzyme site (SpeI in the ’mismatch’ template, BamHI in the ’match’ control) so that amplicon pools could be evaluated for the presence of products derived from either template using gel electrophoresis.

Figure 2 Engineered V3-V4 reporter templates.

Reporter templates were constructed to evaluate the relative performance of primer pairs and the impact of sequence variations in primer binding sites. (A) Schematic of the templates. The match template contained consensus primer-binding sites and the mismatch template contained two sequence changes in the forward and reverse primer-binding sites (G to C at −4 and −8 in the forward binding site; T to G at −14 and G to C at −15 in the reverse binding site). Each template contained a unique restriction site (BamHI in the match, SpeI in the mismatch). (B) qPCR reactions comparing the amplification performance of non-degenerate or degenerate (Pro_341F & Pro_805R) primers using each template independently or an equal mixture of the two (each having the same final template concentration, four replicates each). These reactions used Platinum II Taq polymerase supplemented with EvaGreen and used the EMP protocol. The non-degenerate primers performed better in each case, even on the mismatch template. Poorer performance was associated with higher max/KD ratios. (C) DNA from each qPCR reaction set was pooled, purified, and subsequently digested using either BamHI or SpeI. There was no detectable mismatch amplicon (SpeI digested) in the amplicon pools derived from either of the mixed-template reactions.

When monitored by qPCR, the non-degenerate primers again performed better than the degenerate pair on either template (Fig. 2B). In reactions containing a 1:1 mixture of each template, amplicons derived from the match template dominated using either primer pair and products derived from the mismatched template were undetectable by restriction digestion (Fig. 2C). Therefore, the degenerate primers did not compensate for the template mismatches and they impeded amplicon production from both templates. Taken together, these data suggested that using non-degenerate primers to amplify heterogeneous targets could be a fruitful strategy if the performance on mismatched targets could be improved.

Thermal-bias PCR for co-amplification of a mismatched template

Commonly-employed V3-V4 degenerate primer pairs have predicted annealing Tms that are substantially different (Data SD1). In pilot experiments, it was found that using non-degenerate primers with matched Tms improved the co-amplification of the mismatched reporter to detectable levels (Fig. S4). Following these insights, a new amplification strategy termed “thermal-bias PCR” was developed that uses non-degenerate consensus primers having matched targeting Tms and containing long 5′ tails harboring sequencing adapters and additional nucleotides to substantially raise and match their full-length Tms (Fig. 3A). The thermal-bias protocol employs a low targeting annealing/extension temperature for only the first two cycles (during which the requisite complementary amplicon strands are generated). The cycling protocol then switches to use a very high annealing/extension temperature for the remaining amplification cycles, which forces the reactions to only use amplicons as templates instead of continuously re-targeting the initial templates. Thus, any bias against priming a particular target would only be present for two cycles, with the remainder of the cycles ‘locking in’ relative abundance.

Figure 3 Thermal-bias PCR improves the amplification of a mismatched template.

(A) Schematic of a thermal-bias PCR. Template (green) targeting is performed using two thermal cycles with a low annealing temperature. During targeting, only the 3′ regions of the primers (blue) anneal. The subsequent amplification cycles are performed with a higher temperature that prevents additional targeting, taking advantage of the high Tm of the entire primer. (B) qPCR reactions monitored amplicon production from matched, mismatched, or mixed reporter templates using non-degenerate primers (4 replicates each). These reactions used SsoFast polymerase and contained Illumina sequencing adapters and had matched targeting and amplification Tms (TB_F_Tm55_v1 and TB_R_Tm55_v1: calculated to be ~55 °C and ~79 °C, tail lengths of 44 and 46 nt, respectively). The two targeting cycles were annealed at 48 °C followed by 40 amplification cycles annealed at 78 °C. Similar performance and yields were obtained in each case. (C) DNA from each reaction set was pooled, purified, and digested with SpeI to detect the presence of amplicons derived from the mismatch template. The mismatch amplicon was detectable in the reaction containing equally-mixed templates (lane 4). DNA from the match and mismatch reactions (lanes 1 and 2) was mixed in equal proportions prior to digestion (lane 5), or thermally reannealed prior to digestion to allow mixing of amplicon strands (lane 6). Reannealing lowered the abundance of SpeI digestion fragments to a level similar to that in lane 4, indicating that the mismatch amplicon had been well-represented in the mixed-template reactions. The brightness of the molecular weight marker lane was increased for visibility.

An additional protocol improvement came from the use of an alternative polymerase. SsoFast is a Pfu/Vent hybrid polymerase fused to the Sso7d DNA binding protein. The Sso7d domain not only improves the processivity (extension rate) of polymerases to which it is attached, but it also increases primer annealing Tms (Wu et al., 2017; Wang et al., 2004). Although Taq polymerase functioned in trial thermal-bias protocols, it was found that that SsoFast polymerase tolerated degenerate primers and mismatched templates better (Fig. S5), so it was selected for further studies.

The amplification stages could be performed as high as 80 °C without substantially inhibiting the reactions (the max/KD ratios were similar from 74 °C to 80 °C,) (Fig. S6A). The thermal-bias against re-priming on the initial targets during amplification was confirmed using protocols that lacked the initial targeting steps, which failed to yield amplicons (Fig. S6B). Altering the temperatures or times of the targeting cycles had a negligible impact on performance as long as that temperature was at least ~4 °C lower than the calculated targeting Tm. Thus, a thermal-bias PCR protocol can be employed that has an annealing temperature difference of ~25–30 °C between the targeting and amplification stages.

Using a thermal-bias protocol, amplification of the mismatched V3-V4 template appeared nearly as efficient as the match template and amplicons derived from an even mixture of the two templates contained both versions after 40 PCR cycles (Figs. 3B and 3C). It was noted that qPCR data from reactions using mixed reporter templates exhibited an apparent lag in amplification (e.g., Fig. 3B ‘Mix’), which was puzzling because amplicon strands were presumably being independently primed. The signals generated in those experiments were dependent on a fluorescent dye binding to dsDNA, so one explanation might be that heterogeneously-annealed amplicons (containing a match and mismatch strand) might not have bound as much dye as homogeneous amplicons. Heterogeneous amplicons would also not be digestible by SpeI, which would lower the apparent abundance of the mismatch product. To evaluate this idea, homogeneous match and mismatch amplicons were mixed in equal proportions before digestion. An aliquot of that mixture was then thermally melted and reannealed before each sample was subjected to SpeI digestion (Fig. 3C, lanes 5 and 6). It was evident that reannealing the mixture reduced the abundance of SpeI-digestible amplicons, indicating that heterogeneity in the digestion assays had lowered the abundance estimates.

Thermal-bias PCR accurately reports changes in template abundance

A common goal of amplicon deep-sequencing is to measure changes in the relative abundance of organisms, which requires that invariant populations be consistently reported and that changes in other populations be accurately reported (Morton et al., 2019). To determine the utility of thermal-bias PCR for generating sequencing libraries, the V3-V4 reporter match and mismatch templates were spiked into a commercial mixture of ten bacterial genomes at two different levels (one having half as much spiked as the other). These mixtures were then subjected to a 30-cycle thermal-bias PCR using primers that appended Illumina adapters (Fig. 4A). Replicate samples were pooled, cleaned, and normalized prior bar coding and paired-end sequencing. Sequence reads were then mapped onto a reference sequence that contained each bacterial V3-V4 region as well as the match and mismatch reporter sequences. In this experiment, the ‘reverse’ reads covered the restriction sites in the reporters and allowed them to be unambiguously mapped relative to amplicons derived from E. coli genomes. Read depths were divided by the 16S rRNA gene copy number for each organism and plotted as a percentage of total (Fig. 4B).

Figure 4 Thermal-bias PCR using mixed genome targets.

A commercial mixture containing ten different bacterial genomes was subjected to thermal-bias PCR targeting the V3-V4 region using primers containing Illumina adapters (TB_F_Tm55_v2 and TB_R_Tm55_v2). (A) qPCR of the three library amplifications (same conditions as this figure): Library_1 contained only the bacterial genomes, Library_2 was spiked with a mixture of match and mismatch reporter templates (spike), and Library_3 spiked with half as much mixture (1/2 spiked). (B) Amplicons were pooled and subjected to Illumina sequencing. Sequencing reads were mapped to a reference containing each V3-V4 sequence. Read depths at each position were recorded, corrected for the number of rrs genes in each species, and plotted as a percent of total reads. Species representation was consistent among the three libraries (reads from Hv were also consistent, but too low in abundance to be seen at this plot scale (~0.06%). Although mismatch reporter reads were under-represented in each Library relative to the match reporter reads (64% and 62%), each reporter was represented in Library_3 at ~50% of their abundance in Library_2 (53% and 51%), consistent with the spiking level.

The relative abundances of the genome-derived V3-V4 amplicons was highly similar among the libraries and, for the most part, aligned with several independently measured genome compositions of this mixture (Foox et al., 2021). Nine of these genomes had consensus primer-binding sites and the relative abundances were not correlated with the G/C content of their V3-V4 regions. One notable exception was a substantial underrepresentation of the Archaeon H. volcanii, which has a high G/C content in its genome, but not in the V3-V4 region. This underrepresentation was unexpected because the primer-binding sites of H. volcanii presented three mismatches: two were the same as those in the engineered mismatch template (G to C at −8 in the forward site, and T to G at −14 in the reverse), with the third H. volcanii mismatch being outside of a 3′ region (A to G at −9 in the forward site). This V3-V4 region has as other properties that likely contributed to an inefficient targeting stage (discussed below).

Both the match and mismatch reporter amplicons were present in the spiked libraries, but the mismatch product was reduced by ~37% compared to the match product in each case. Importantly, each reporter version was reduced in abundance by ~50% in Library_3 compared to Library_2, accurately reflecting the difference in spiking level. Taken together, any biases in the targeting stages were preserved through the amplification stages along with apparent differences in relative template abundance. Therefore, the thermal-bias protocol is a promising alternative method for generating sequencing libraries and it can tolerate non-consensus targets.

Discussion

The experiments presented here demonstrate the utility of using the max/KD ratio as a convenient metric to assess reaction performance that can be applied to any scenario where comparisons are being made between samples or between reaction conditions. Because each reaction exhibits unique amplification behaviors, changes in max/KD have so far been subjectively evaluated by monitoring variances among replicates. Additional characterizations of the relationship between max/KD and reaction efficiency are ongoing, with a goal of developing an objective metric that will allow the quality of each reaction to be statistically weighted during replicate averaging.

It is known that degenerate primers can perform poorly when comparing amplicon yields (Gaby & Buckley, 2017; Bonk et al., 2018), yet systematic evaluations of their influence on reaction quality has not been tractable. The presented data indicate that primer degeneracy imparted undesirable side effects by dampening amplicon production from both canonical and non-canonical targets. Moreover, they did not compensate for target mismatches better than non-degenerate versions. Modeling simulations revealed that elevations in the max/KD ratio are indicative of reaction poisoning, which suggests that certain classes of annealed mismatched oligonucleotides do not serve as primers and they likely block target sites from receiving productive variants. That being said, there are obvious scenarios where degenerate primers are preferred, such as amplifying DNA homologs containing unknown sequence variations.

There are numerous reports of factors that contribute to PCR bias and several innovative approaches have been developed to reduce bias during library formation. Of those, the ‘deconstructed PCR’ (DePCR) methods are most similar to thermal-bias PCR (Green, Venkatramanan & Naqib, 2015; Naqib, Poggi & Green, 2019; Naqib et al., 2019; Kahsen et al., 2024). In DePCR protocols, the initial targeting stages are performed at low annealing temperatures for only two cycles using tailed degenerate primers, and the resulting products are purified prior to being used as templates for a separate amplification stage using different non-degenerate primers that anneal to the targeting primer tails. During DePCR development, it was observed that low annealing temperatures strongly biased primer selection and that certain primer sequences appeared to be rejected (Kahsen et al., 2024). Those observations are consistent with the finding here that PCR reactions were inhibited by annealed degenerate oligonucleotides that failed to prime.

Primer extension rates of DNA polymerases can vary widely depending on processivity and incubation temperature. During early characterizations of Taq polymerase, it was found that it has an optimal extension temperature of 80 °C in vitro (Chien, Edgar & Trela, 1976; Lawyer et al., 1993). However, for its use in PCR, the initial extension temperature has to be low enough to promote short primer annealing (Saiki et al., 1988). Modern protocols optimized for yield elevate the temperature after the initial annealing step to provide a balance between primer-template stability and rapid elongation. Other unavoidable factors that reduce PCR performance include repeated re-priming of the initial targets, which causes a linear accumulation of DNA that is dependent on the concentration of initial template (Green, Venkatramanan & Naqib, 2015). Also, structural interference by stem-loops in the template impede polymerase progression, and such structures are more stable at lower temperatures (Fan et al., 2019). Although each of these issues are largely alleviated by thermal-bias PCR, it should be noted that the sequence between primer binding sites may still substantially impact performance. For example, at the targeting temperature used to create the sequencing library for Fig. 4 (48 °C), the calculated ΔG of folding for a collection of stem-loop structures in the H. volcanii V3-V4 region is twice that found for the other bacteria (−49.6 kcal/mol vs. an average of −24.4 +/− 3.2 kcal/mol, computed using 250 mM Na+ and 5 mM Mg++) (Zuker, 2003). In addition, the Hv forward primer binding site is predicted to contain a four base pair stem-loop that is not present in the other species. Because the mismatch reporter template contained four mismatches whereas the Hv target contained three (two in common with the mismatch reporter), these secondary structures likely contributed to the poor representation of this species in the library.

Conclusions

Thermal-bias PCR is a simple, affordable, one-reaction tool that can be used for qPCR or library generation. For routine library preparations, it provides a balance between absolute representation and sensitivity to changes in community structure. Iterations of the protocol can include adding barcodes to the primers or using alternative adapter sequences (Saiki et al., 1988; Hamady et al., 2008). This study employed an engineered template to investigate the ability of commonly used degenerate primers to compensate for sequence variations at those prescribed positions, and also to evaluate the utility of the thermal-bias protocol. However, it should be emphasized that targets containing mismatches in the 3′ region of a primer binding site are likely to be poorly amplified by nondegenerate primers. Although it was beyond the scope of this project, having a direct comparison between libraries generated using conventional degenerate primers and libraries prepared using thermal-bias PCR would shed more light on the impact of degenerate primer interference and their ability to overcome 3′ mismatches. Proofreading polymerases offer a solution to 3′ mismatches and allow for the efficient amplification of non-canonical targets (Gohl et al., 2021). Such editing introduces changes to the amplicons that must be overcome by additional editing in each subsequent cycle, so it will be exciting to see if future studies reveal that a combination of these approaches can further improve amplicon balance.

Supplemental Information

Supplemental Information 1 Modeling the influence of max and KD on PCR reaction quality.

The PCR model was used to generate reaction profiles using different input values. (A) The PCR model: the amplicon yield in a given thermal cycle is a function of the prior cycle’s amplicon abundance (prev), the maximum theoretical yield of the completed reaction (max), and the inhibitory activity of accumulated products (KD). After a reaction’s max and KD values are obtained by fitting the model to qPCR data, the abundance of the initial template can be calculated (seed). (B) Modeled reaction profiles using different seed values of 0.1, 0.001, or 0.00001 and with invariant max = 1E4 and KD = 1E3. The different seed values shift the positions of the reaction profiles, but the profile shapes remain the same. (C) Using invariant seed, max, and KD values, the maximum possible per-cycle efficiency was set to 100% (2-fold amplification), 95% (1.9-fold amplification), or 90% (1.8-fold amplification). (D) Reactions in which the value of max was reduced in 2-fold increments. (E) Reactions in which the value of KD was reduced in 2-fold increments. (F) Different ratios of max/KD change the reaction profiles and can produce the same yield at a given cycle. Shown are three examples that produce the same yield at the 30th cycle (marked in red): max = 1E4, KD = 1E3 (max/KD = 10); max = 1.83E4, KD = 5E2 (max/KD = 36.6); and max = 7.8E4, KD = 2.5E2 (max/KD = 312).

Supplemental Information 2 Primer sequences and Tm calculations.

Oligonucleotide primer sequences and calculated Tms.

Supplemental Information 3 Different polymerases using degenerate or non-degenerate primers.

The V3-V4 region was amplified from E. coli genomic DNA using reaction conditions recommended by the enzyme manufactures. Top panel, SsoFast EvaGreen (Bio-Rad); middle panel, iTaq SYBR Green (Bio-Rad); bottom panel, Q5 (NEB) supplemented with EvaGreen (Jena Bioscience).

Supplemental Information 4 Engineered V3-V4 reporter templates.

Shown are the nucleotide sequences of the top strands of the ’match’ and ’mismatch’ reporters. Each has a unique restriction site inserted. The mismatch version has four nucleotide changes (red) in the primer-binding regions that are in positions that should be compensated by the degenerate primers.

Supplemental Information 5 Matching Tms improve production of a mismatch amplicon.

Non-degenerate primer pairs were evaluated for their ability to co-amplify the mismatch V3-V4 template (MM) in the presence of an equal amount of the match template (M). The non-degenerate pair ND_F1 and ND_F2_Tm54 and ND_R1 had matching Tms of ~54 °C (left set, lanes 2-4); the non-degenerate pair ND_F1 and ND_R2_Tm57 had matching Tms of ~57 °C (right set, lanes 5-7). PCRs used Platinum II Taq polymerase and the EMP cycling protocol. Purified PCR products were digested with SpeI to detect the mismatch amplicon. The mismatch amplicon was detected in the mixed template reactions (lanes 4 and 6).

Supplemental Information 6 Degenerate tolerance and thermal-bias PCR performance.

(A) SsoFast amplification of V3-V4 from E. coli genomic DNA (1.56 μg/μL) using nondegenerate or degenerate primers (ND_F1/R1 or Pro_341F/Pro_805R); 50 °C annealing/extension 60 sec. (B) Thermal-bias PCR using either SsoFast or Platinum Taq with either a match or mismatch engineered template (primers TB_F_Tm55_v1/TB_R_Tm55_v1). The Platinum Taq reactions were supplmented with EvaGreen. Targeting cycles used 50 °C annealing/extension for 5 min, amplification cycles used 80 °C annealing/extension for 1 min.

Supplemental Information 7 Amplification and target-priming thermal-bias.

A) The match and mismatch V3-V4 templates were used to evaluate PCR performance at various elevated amplification temperatures. Two targeting cycles were performed at 50 °C and 40 amplification cycles were performed at the indicated annealing temperatures. B) The match template was used in a PCR protocol lacking the two targeting cycles, but contained 30 amplification cycles at the indicated temperatures. The absence of targeting strongly suppressed amplification at 74 °C (~11 cycles, ~2000-fold), and abolished amplification at 78 °C and 80 °C. These experiments used SsoFast polymerase and the primer targeting sections had calculated Tms of 55 °C and they contained Illumina TruSeq adapter sequences in their tails (TB_F_Tm55_v2 and TB_R_Tm55_v2).

Supplemental Information 8 Uncropped agarose gel images.

Full size gel images of figure panels 2C, 3C, and S4.

Supplemental Information 9 Raw qPCR fluorescence and global fitting values.

Index, raw fluorescence data, and global fitting analyses.

Supplemental Information 10 MIQE checklist for real-time PCR analysis from Bustin et al., 2009.

The author thanks Jon Caranto, Herve Roy, and Hubert Salvail for their editorial comments. Undergraduate students participating in the Burnett School’s Applied Industrial Microbiology program contributed to pilot studies.

Additional Information and Declarations

Competing Interests

The author declares that he has no competing interests.

Author Contributions

Sean D. Moore conceived and designed the experiments, performed the experiments, analyzed the data, prepared figures and/or tables, authored or reviewed drafts of the article, and approved the final draft.

DNA Deposition

The following information was supplied regarding the deposition of DNA sequences:

The Illumina FASTQ sequences are available at the NCBI Sequence Read Archive: BioProject PRJNA1258214.

Data Availability

The following information was supplied regarding data availability:

The raw data is available in the Supplemental Files.

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
