# Peer review of "Thermal-bias PCR: generation of amplicon libraries without degenerate primer interference"

_PeerJ, doi:10.7717/peerj.20241_

## Round 0.1 · original submission · Major Revisions

· Academic Editor

Major Revisions

Please address all the comments and submit a revised version.

Reviewer 1 ·

Basic reporting

The manuscript by Moore focuses on the use of alternatives to degenerate primers to amplify homologous genes across multiple species. Through both modeling and experimental data, the authors demonstrate a key limitation of degenerate primers: they hinder the efficient synthesis of the initial product pool. While the idea that non-degenerate primers more effectively amplify target sequences compared to their degenerate counterparts may seem intuitive, it warrants a more thorough explanation. Notably, the authors ultimately use non-degenerate primers in their experiments with great success using a “thermal-bias protocol” protocol. The data are clearly presented, and in this reviewer’s opinion, the manuscript merits publication in PeerJ.
Specific comments
Introduction:
1.- The paper would benefit from a schematic representation of the problem. For example, the consequences of using degenerate primers at the 5′ end differ from those of using them at the 3′ end. DNA polymerase is particularly inefficient at extending from a mismatched 3′ end, which can significantly affect amplification efficiency.

Results
1. Figure 1:
The data presented in Figure 1 show qPCR experiments using both non-degenerate and degenerate primer pairs targeting E. coli rrs variable regions. As expected, the non-degenerate primers exhibit robust amplification compared to the degenerate pairs. The authors should emphasize that DNA polymerases are particularly inefficient at extending from mismatches located near the 3′ end, in contrast to mismatches occurring 4–5 nucleotides upstream. A schematic illustrating this concept, along with a clear description of the primers used, would greatly improve clarity.
2. Figure 2:
The data in Figure 2 have an important caveat: the reviewer cannot determine the distance between the mismatches and the 3′ end of the primers. Numerous studies have shown that DNA polymerases poorly tolerate mismatches near the 3′ end, including the seminal paper by the group of Lorena Beese:
Johnson & Beese, Cell. 2004 Mar 19;116(6):803–816. doi: 10.1016/s0092-8674(04)00252-1
“Structures of mismatch replication errors observed in a DNA polymerase”
The authors should consider referencing this and clarifying mismatch positioning in their figure legend or methods section.
3. Figure 3:
In this figure, the authors employ non-degenerate consensus primers with extended 5′ tails and a novel PCR protocol using low annealing and extension temperatures. The most significant contribution of the manuscript (from the humble point of view of this reviewer) is the development of this "thermal-bias" protocol, which enables efficient amplification of both matched and mismatched templates—an impressive achievement. Two minor suggestions:
a) Please specify the length of the 5′ tails used.
b) Consider requesting permission from PeerJ staff to enhance the visibility of the molecular weight markers in the gel image, and include base-pair sizes for reference.
Regarding the statement:
"Although Taq polymerase functioned in trial thermal-bias protocols, it was found that SsoFast polymerase tolerated degenerate primers and mismatched templates better, so it was selected for further studies.”
The authors may consider including this comparison data as a supplementary figure.
4. Figure 4:
Here, the thermal-bias protocol is used to amplify V3–V4 regions from various species. The authors successfully amplify products in all tested organisms, but the archaeon H. volcanii, demonstrating the robustness and broad applicability of the approach.

Experimental design

the experimental design is well executed..no comment

Validity of the findings

the analysis and validation of the data follows stablished protocols..no comment

Additional comments

None

Reviewer 2 ·

Basic reporting

-rrs is not the proper nomenclature for the 16S rRNA locus (I believe that rrs is more commonly used to refer to a replication control element). I would recommend replacing this abbreviation with "16S rRNA gene" throughout.
-Line 398: "bar codes" should be one word ("barcodes").

Experimental design

-Lines 252-254. Since mismatches near the 3' end of the primers typically have a higher amplification penalty (see Bru et al, 2008 Applied Environmental Microbiology; Wu et al, 2009, Journal of Microbiology Methods; and Gohl et al, 2021 Nucleic Acids Research), it should be noted that these internal mismatches to the degenerate portions V3V4 primers are not the most extreme hypothetical mismatch scenario.
-Line 330. The statement "Nine of these genomes had consensus primer-binding sites" needs clarification. Does this mean that 9/10 had perfect matches to the V3-V4 primers and just the H. volcanii strain had mismatches? If so, this doesn't represent a very compelling test of the effectiveness of the thermal bias PCR in successfully amplifying mismatched templates in moderately complex mixtures (since the only mismatched sequence, excepting the spiked amplicon mismatch reporter template) was not amplified).
-The paper would be substantially strengthened with additional tests of more complex samples, containing a larger number of potential primer mismatches to the V3-V4 targeting sequences in the thermal-bias PCR primers.
-In Figure 4, it would be desirable to do a comparison of the mock community thermal-bias PCR results with a standard V3-V4 amplification (using degenerate primers).

Validity of the findings

This paper addresses a legitimate issue with multi-template PCR reactions and demonstrates both a downside to using degenerate primers and a potential mitigation strategy. The experiments presented in the paper and clear and well controlled and the proposed thermal-bias PCR strategy makes good sense in theory, but the tests of the ability of thermal-bias PCR are limited in their scope and consist of just:
1) A single pair of match/mismatch templates constructed from the E. coli 16S V3-V4 locus.
2) A mock community sample for which the majority of organisms seem to match the non-degenerate primer (with the one organism with mismatches being severely under-represented in the sequencing data).
In order to really substantiate that this approach is a viable strategy for overcoming the limitations of degenerate primers, tests on additional more complex samples would be desirable (both mock communities with a larger number of organisms with predicted primer mismatches and actual microbiome samples, with comparisons to data generated with degenerate primers and/or shotgun sequencing data). Alternatively, the conclusions of the paper could be dialed back to acknowledge that more substantial testing is needed to confirm that thermal-bias PCR is effective in amplifying a broad range of mismatched templates (as currently there is just a single successful example: the mismatch reporter template versus an unsuccessful example: the H. volcanii strain in the mock community).

Additional comments

"Thermal-bias PCR: generation of amplicon libraries without degenerate primer interference" describes an approach to improve the amplification of mixed template samples, such as those commonly encountered in microbiome profiling. The author has previously developed a baseline-independent amplification curve fitting model for estimating template abundance in qPCR. In this study, parameters from this curve-fitting model are used to assess the quality of amplification reactions and to compare degenerate and non-degenerate primers. Primers with degeneracies are shown to amplify less efficiently than non-degenerate primers (Figs. 1-2), but the non-degenerate primers show a bias towards perfectly matched templates using standard PCR cycling conditions (Fig. 2).

The author then proposes the use of "thermal-bias" PCR as a means to achieve proportional amplification in mixed template PCR reactions without relying on degenerate primers. The thermal-bias PCR involves two initial cycles at a permissive annealing temp, with Tm-matched non-degenerate primers containing a sequence tail, followed by many cycles of amplification with a much higher, stringent annealing temp, with the goal of amplifying any amplicons tagged with the primers in the first two PCR cycles. The thermal-bias PCR approach improves amplification efficiency, while enabling representation of templates with primer mismatches in a simple (1:1) mixture of constructed reporter templates. The thermal-bias PCR workflow was then tested on a microbial DNA mock community sample that was either not spiked or spiked with different amounts of mismatch reporter template.

---

## Round 0.2 · accepted · Accept

· Academic Editor

Accept

Congratulations, your manuscriot is accepted!

Reviewer 1 ·

Basic reporting

I this revised version of the manuscript the author addressed all my previous concerns/comments. In the opinion of this reviewer the manuscript is suitable for it acceptance.

Experimental design

No new comments.

Validity of the findings

No new comments.

Additional comments

No new comments.

Reviewer 2 ·

Basic reporting

no comment

Experimental design

no comment

Validity of the findings

no comment

Additional comments

The revised manuscript addresses the key concerns of the reviewers. While more experiments will be useful to fully characterize the performance of thermal-bias PCR relative to the incorporation of degeneracies, the author acknowledges this in the text and suggests areas for future work. Also, apologies for previously being mistaken regarding the rrs nomenclature. The author should feel free to reverse these changes as desired in the final version.